# Bearing-DETR: A Lightweight Deep Learning Model for Bearing Defect Detection Based on RT-DETR

**DOI:** 10.3390/s24134262

**Published:** 2024-06-30

**Authors:** Minggao Liu, Haifeng Wang, Luyao Du, Fangsong Ji, Ming Zhang

**Affiliations:** 1School of Energy and Mining Engineering, Shandong University of Science and Technology, Qingdao 266590, China; 202283010090@sdust.edu.cn (M.L.); 202383010015@sdust.edu.cn (F.J.); 2School of Information Science and Engineering, Linyi University, Linyi 276002, China; wanghaifeng@lyu.edu.cn; 3School of Automation, Wuhan University of Technology, Wuhan 430070, China; duluyao@whut.edu.cn

**Keywords:** bearing defect detection, real-time detection transformer (RT-DETR), lightweight model, industrial efficiency

## Abstract

Detecting bearing defects accurately and efficiently is critical for industrial safety and efficiency. This paper introduces Bearing-DETR, a deep learning model optimised using the Real-Time Detection Transformer (RT-DETR) architecture. Enhanced with Dysample Dynamic Upsampling, Efficient Model Optimization (EMO) with Meta-Mobile Blocks (MMB), and Deformable Large Kernel Attention (D-LKA), Bearing-DETR offers significant improvements in defect detection while maintaining a lightweight framework suitable for low-resource devices. Validated on a dataset from a chemical plant, Bearing-DETR outperformed the standard RT-DETR, achieving a mean average precision (mAP) of 94.3% at IoU = 0.5 and 57.5% at IoU = 0.5–0.95. It also reduced floating-point operations (FLOPs) to 8.2 G and parameters to 3.2 M, underscoring its enhanced efficiency and reduced computational demands. These results demonstrate the potential of Bearing-DETR to transform maintenance strategies and quality control across manufacturing environments, emphasising adaptability and impact on sustainability and operational costs.

## 1. Introduction

Bearings are indispensable components in chemical equipment, serving as crucial supports for rotating shafts. Their performance profoundly impacts equipment stability and reliability. Inadequate bearing functioning can lead to imbalances in rotating parts, resulting in heightened vibration and noise and, in severe cases, equipment failure or damage. Such failures pose significant mechanical hazards, potentially causing accidents, injuries, and property damage. Moreover, bearing maintenance and replacement necessitate equipment suspension, reducing production efficiency. Hence, ensuring the seamless operation of the chemical production process hinges on the reliability of bearings.

Various defects, such as grooves, wear, and scratches, can compromise the quality of bearings during production, assembly, or transportation. Detecting these defects is paramount to maintaining equipment functionality and safety. While traditional defect detection methods, like visual inspection [1] and simple sensor-based techniques [2], offer partial solutions, they often fall short in meeting the demands of high precision and real-time monitoring. The emergence of computer vision and AI presents novel avenues for bearing defect detection. These modern approaches offer enhanced intelligence, accuracy, and efficiency, ushering in breakthroughs in defect detection.

In this context, the development of lightweight analysis models holds significant importance. Lightweight models, such as RT-DETR [3], offer a streamlined approach to defect detection, balancing performance with computational efficiency. These models enhance defect detection capabilities by leveraging deep learning techniques while minimising computational overhead. This convergence of advanced AI methodologies with lightweight design principles represents a pivotal advancement in bearing defect detection, promising heightened efficiency and accuracy in industrial applications.

Object detection, a crucial computer vision task, entails locating and identifying objects in images or videos. Over the decades, researchers have crafted myriad detection methodologies, ranging from feature-based to template-based and, more recently, deep learning approaches. Among these, deep learning, especially Convolutional Neural Networks (CNNs) [4], has seen remarkable strides, with models like Faster R-CNN [5], YOLO [6], and SSD [6] achieving impressive results.

Figure 1 shows the evolution of significant object detection models, highlighting the progression from traditional methods to advanced deep learning techniques. Figure 1, titled “Timeline of YOLO and DETR Series Developments in Object Detection Algorithms”, visually represents this technological evolution.

In the lineage of these developments, as illustrated in Figure 1, RT-DETR represents a synthesis of two influential object detection paradigms: the Transformer and DETR (Detection Transformer). The Transformer, initially conceived for natural language processing, has proven exceedingly efficacious in computer vision tasks. DETR, an end-to-end object detection model, re-envisions the detection task as an object query problem adeptly addressed using Transformer architecture. RT-DETR inherits the architectural innovations of DETR but introduces optimisations that cater to real-time object detection, making it a highly relevant and timely solution in scenarios demanding swift and accurate detection.

Our research aims to capitalise on the real-time processing capabilities of RT-DETR while tailoring the model to the nuanced requirements of detecting defects in industrial bearings—a challenge that necessitates not only the precise localisation of defects but also the capability to process and analyse images swiftly to minimise downtime and enhance maintenance efficiency. Through this lens, Bearing-DETR emerges as a specialised, lightweight model that embodies the cutting-edge RT-DETR approach, promising a new frontier in computer vision application to industrial quality control. The specific contributions of this work are as follows:

(1) Incorporating the Dysample technique within the RT-DETR algorithm substantially streamlines the upsampling process. By simplifying the architecture in line with VanillaNet principles, Dysample aids RT-DETR in reducing computational overhead and model complexity. This facilitates the detection of bearing defects with increased accuracy and speed, ensuring that RT-DETR remains both lightweight for real-time applications and effective in identifying nuanced defect features while optimising computational resource use.

(2) Efficient Model Optimization (EMO) introduces the Meta-Mobile Block (MMB), enhancing RT-DETR with a streamlined, scalable architecture. Merging IRBs with Transformer elements equips RT-DETR with adaptability for real-time defect detection, optimising performance while reducing complexity and computational resources. This innovative approach refines RT-DETR efficiency, making it a formidable tool for the precision-oriented task of bearing defect detection.

(3) The Deformable Large Kernel Attention (D-LKA) incorporated in RT-DETR brings advancements to object detection. D-LKA utilises expansive kernels and deformable convolutions to efficiently process data with flexibility and depth, which is crucial for high-resolution tasks. In RT-DETR, D-LKA aids in effectively interpreting complex patterns in imagery, which is vital for accurate bearing defect detection. This integration improves the capability of RT-DETR to analyse intricate visual information without incurring high computational costs, showcasing the potential for industrial applications beyond medical image segmentation.

In response to the urgent need for improved bearing defect detection, this research introduces the innovative Bearing-DETR model, leveraging the RT-DETR framework with three key enhancements: Dysample dynamic upsampling for streamlined network efficiency, Efficient Model Optimization (EMO) for robust feature extraction with minimal computational overhead, and Deformable Large Kernel Attention (D-LKA) for precise defect localisation even in complex imaging scenarios. These advances significantly boost the ability of RT-DETR to detect subtle and critical bearing defects that could affect production stability and safety. Bearing-DETR achieves high detection precision, enhancing real-time monitoring and ensuring equipment reliability in industrial settings.

This is divided into the following sections: Section 2 provides an overview of the current state of research in the field of defect recognition, both domestically and internationally. Section 3 primarily introduces RT-DETR, and the methods developed in this work to enhance it are referred to as Bearing-DETR. Section 4 focuses on various experiments and their results. Section 5 describes the conclusions and prospects of this work.

## 2. Related Work

### 2.1. Bearing Detection

Reviewing traditional methods such as vibration analysis and acoustic emission monitoring is essential to understanding the evolution of bearing defect detection. These techniques were foundational in early industrial applications and are renowned for their efficacy and widespread use.

Vibration analysis has been a cornerstone in mechanical fault detection, with significant enhancements over time. A study by Minmin Xu et al. [7] developed a sophisticated nonlinear vibration model with fourteen degrees of freedom. This model effectively captures the dynamics of gear meshing forces and variations in radial clearance. The introduction of the Modulation Signal Bispectrum-Sideband Estimator (MSB-SE) for monitoring bearing clearance changes marks a substantial improvement over traditional methods like BPFO amplitude analysis, RMS, and kurtosis, demonstrating superior accuracy in fault diagnosis.

Addressing the limitations of relying solely on vibration data, Josué Pacheco-Chérrez [8] advocates for a combined approach using both acoustic and vibration signals tailored to the specific failure types and signal frequencies. This integration enhances precision in fault detection, with his research achieving fault classification accuracies above 96% using machine learning techniques, showcasing the potential of combined non-invasive methods in predictive maintenance strategies.

Exploring innovative noninvasive methods, Anurag Choudhary [9] introduces thermal imaging as a tool for diagnosing bearing faults. Thermal imaging effectively identifies faults under various conditions by employing both shallow and deep learning techniques, specifically an Artificial Neural Network (ANN) and a Convolutional Neural Network (CNN) based on the LeNet-5 architecture. This method offers early warnings and significantly reduces system downtime, underscoring its utility in predictive maintenance.

Furthermore, Chunmin Yu [10] focuses on the design and optimisation of three degrees-of-freedom hybrid magnetic bearings (3-DOF HMBs) for high-speed applications. His research introduces a dynamic magnetic circuit model that incorporates eddy current and saturation effects, proposing a multi-objective optimisation design method that significantly enhances the performance and cost-efficiency of these bearings.

While foundational and widely used, traditional bearing detection methods encounter several challenges in modern applications. They often have limited data processing capabilities, which hinder real-time monitoring and rapid fault diagnosis. Many conventional techniques also struggle with handling complex or ambiguous image data, reducing detection accuracies in complex industrial environments. Furthermore, these methods typically require extensive historical data for accurate fault prediction, which can delay diagnoses of new or uncommon defects. Additionally, traditional approaches are sensitive to environmental interference, such as noise and operational variations, which can affect their reliability, leading to potential false positives or missed detections.

These inherent limitations underscore the need for more advanced and versatile detection methods. The shift towards integrating technologies like deep learning, thermal imaging, and optimised magnetic bearings aims to overcome these challenges, setting new standards for efficiency and reliability in bearing fault diagnostics.

Deep learning technologies, particularly convolutional neural networks (CNNs), have transformed the object detection landscape by overcoming the inherent limitations of traditional methods. Zhao et al. [11] thoroughly review the evolution from manual feature crafting to automated feature extraction through advanced neural architectures. Their research underscores the transition to deep learning, which facilitates the extraction of semantic and high-level features, enhancing both the efficiency and effectiveness of detection systems.

The comprehensive analysis by Zhao highlights how CNNs significantly outperform traditional object detection approaches through distinct network architectures and training strategies. This technological leap addresses the stagnation in performance previously experienced due to the reliance on complex ensembles that blended multiple low-level image features with high-level context from object detectors and scene classifiers.

Building on the foundation of Zhao et al., Du et al. [12] delve into two-stage object detection algorithms, examining their current status and detailing the mechanisms of prominent algorithms like Faster R-CNN and Cascade R-CNN. Their work illuminates the capabilities of these systems to refine detections through sequential processing stages, enhancing the accuracy of the final output.

In a practical application of two-stage detection, Wang et al. [5] implement an enhanced version of Faster R-CNN to detect wear status in mechanical equipment. By moving away from traditional high-resolution imaging methods, which are not only time-consuming but also rely heavily on subjective interpretation, Wang’s method provides a more objective and efficient solution. The use of this advanced model in a real-world setting demonstrates a significant improvement, with detection accuracy surpassing 99%.

Further extending the application of two-stage detectors, Peng et al. [13] introduce a novel bearing defect detection network that integrates a deformable ResNet50 with a double attention feature pyramid network. This system handles the challenges of detecting defects amid random shapes and multiscale artefacts, significantly improving feature extraction accuracy.

Chen et al. [14] address a critical gap in two-stage detection processes by developing the Multi-Path Detection Calibration Network (PDC-Net). Their innovative approach tackles the often-overlooked discrepancies in data distribution between object proposals and refined bounding boxes, enhancing overall detection precision.

Chai et al. [15] adapt the Cascade R-CNN framework to enhance Synthetic Aperture Radar (SAR) image analysis for detecting small-sized ship targets against complex backgrounds. Their modifications, which include the integration of advanced modules like Res2Net and a new focal loss function, yield substantial improvements in detection capabilities.

Hou et al. [16] further refine the application of Cascade R-CNN in remote sensing imagery with the development of Parallel Cascade R-CNN. This novel architecture, designed to handle the diversity of scales and orientations typical in aerial images, demonstrates the potential for tailored detection strategies that improve overall performance metrics significantly.

These studies collectively showcase the robust capabilities of two-stage detection systems in handling complex detection tasks across various domains. However, the single-stage detection methods, notably the YOLO series, bring their advantages, especially in scenarios where speed and efficiency are paramount.

The YOLO (You Only Look Once) series, known for its single-stage detection process, eliminates the proposal stage, streamlining the detection process. This approach speeds up the detection and maintains high efficiency, making it suitable for applications where real-time detection is crucial.

Zheng et al. [17] utilise an improved YOLOv3 model to tackle defect detection in industrial settings, explicitly focusing on bearing cover defects. Their enhancements to the YOLOv3 model allow for better handling of larger defect targets and more effective detection of subtle defects, which are critical in maintaining mechanical systems’ health and operational efficiency.

Merainani et al. [18] apply a modified YOLO-v4 to detect overheated components on rail-road cars, enhancing railway safety through infrared thermal image sequences. This application highlights the adaptability of YOLO architectures to specialised imaging techniques, providing reliable detection in critical safety applications.

Continuing from the developments by Zhao et al. [19], who enhanced the reliability of defect detection in chemical transmission equipment, Jiamin Tao et al. [20] also contribute significantly to the field with their application of advanced imaging and deep learning techniques. They developed the YOLO-OurNet to detect surface defects such as shallow dents and rust on drum-shaped rollers. Their novel approach leverages the fringe projection technique to enhance the visibility of defects, significantly improving detection precision. This method not only overcomes the limitations of traditional visual inspection methods but also highlights the utility of deep learning in refining defect detection under challenging conditions.

Wu Yukun et al. [21] introduce the innovative G-YOLO model for diagnosing faults in rolling bearings. They achieve a high level of diagnostic accuracy by transforming time-domain vibration signals into feature images, which are then analysed using the G-YOLO model. This method stands out for its ability to identify subtle variations in vibrational patterns associated with different bearing conditions, thereby significantly enhancing the reliability of fault diagnostics in industrial machinery.

Finally, Minggao Liu et al. [22] present the YOLOv8-LMG, a state-of-the-art model for detecting bearing defects in industrial machinery. Building upon the YOLOv8n framework, this model incorporates several advanced technologies to enhance detection performance. With features like the VanillaNet Backbone Network and the Lion Optimizer, the YOLOv8-LMG achieves high detection accuracy. It maintains low computational complexity, making it ideal for real-time applications in industrial settings.

These advanced applications of single-stage detectors like the YOLO series illustrate the significant strides in the field. While the YOLO architecture offers a swift and efficient detection process suitable for real-time applications, the continuous innovations and refinements further enhance its accuracy and utility across diverse industrial applications.

Despite the remarkable advancements in two-stage and single-stage detection technologies, deep learning models still face significant challenges, such as the need for large volumes of training data, vulnerability to overfitting, and intensive computational requirements. To address these issues, future research focuses on innovative training strategies, model optimisation techniques, and integrating semi-supervised and unsupervised learning approaches. These efforts aim to reduce the dependency on large labelled datasets and improve the generalisation capabilities of the models, thereby broadening their applicability and effectiveness in real-world scenarios.

In conclusion, the ongoing evolution of object detection methodologies, mainly through deep learning techniques, significantly enhances the precision and efficiency of defect detection systems. These technologies are not only reshaping the landscape of industrial quality control. Still, they are also setting new standards for reliability and performance, ensuring safer and more efficient operations across various sectors.

### 2.2. Detection Transformer

The RT-DETR framework, a hybrid of transformer architectures and convolutional neural networks (CNNs), is at the forefront of the evolution in object detection technologies. This model translates complex images into an array of features efficiently parsed by the transformer to deliver precise object classifications and locations.

Zhu et al. [23] have successfully tailored RT-DETR to enhance uncrewed aerial vehicle (UAV) target detection, thereby significantly improving accuracy and operational efficiency—critical factors for UAV navigation and surveillance reliability. Innovations such as Dilated Re-param Blocks and Gather-and-Distribute Mechanisms showcase customised solutions aimed at optimising the fusion of multi-scale features, which in turn boosts the UAV detection capabilities of the model.

In the medical field, Guemas [24] applies RT-DETR to detect Plasmodium species from thin blood smears. This novel application highlights RT-DETR’s potential to revolutionise diagnostics in global health, particularly for malaria, enabling rapid and accurate parasite identification using easily accessible technology.

The broad adaptability of RT-DETR across various domains—from traffic control to wildlife monitoring and sophisticated surveillance systems—demonstrates its robustness and flexibility. However, despite these successful applications, it is essential to note that RT-DETR technology is still relatively new. Its adoption is not yet widespread, so the volume of literature and research dedicated explicitly to RT-DETR is somewhat limited. This nascent stage presents a unique opportunity for pioneering research and application in unexplored fields.

The foundational aspects of RT-DETR have spurred several essential modifications, enhancing both performance and operational efficiency:

1. Conditional DETR: Muzammul [25] explores this modification, introducing a conditional spatial query mechanism, reducing training complexity and hastening convergence times—features that enhance real-time processing capabilities.

2. Deformable DETR: Liu [26] investigates the integration of deformable attention modules, allowing the model to adaptively concentrate on specific areas based on objects’ size and shape. This capability mainly benefits cluttered scenes, improving the model’s detection accuracy and flexibility.

These enhancements solidify the effectiveness of RT-DETR in traditional applications and extend its utility into new areas such as industrial defect detection and medical diagnostics.

Given the significant advances in RT-DETR, its application in detecting bearing defects holds considerable promise. The Conditional DETR and Deformable DETR could revolutionise quality assurance in manufacturing by providing precise, real-time detection of bearing anomalies, which is crucial for maintaining the integrity of industrial operations. The capability of these models to process complex visual information rapidly and accurately could markedly improve automated monitoring and maintenance systems.

The path forward for RT-DETR involves further refinement to meet specific challenges in industrial applications, particularly bearing defect detection:

(1) Customization for specific industrial needs: Enhancing models to specifically tackle unique industrial challenges, such as varying noise levels and the need for precise, high-resolution detection capabilities.

(2) Integration with other sensory technologies: Combining RT-DETR with diverse sensor inputs creates a multifaceted diagnostic tool, thus enhancing its effectiveness and reliability.

(3) Computational efficiency improvements: Developing computationally less demanding algorithms to enable broader deployment on edge devices and facilitate real-time monitoring across various industrial settings.

In sum, the ongoing development of RT-DETR is poised to lead breakthroughs in detection technologies, potentially redefining automated monitoring and maintenance practices across multiple industries. The emerging nature of RT-DETR technology offers fertile ground for groundbreaking research and application, promising substantial advancements in the accuracy, speed, and reliability of critical detection tasks.

## 3. Algorithm

### 3.1. RT-DETR

In January 2023, Baidu Inc. introduced a significant advancement in real-time object detection frameworks with the release of RT-DETR. This model stands out for its ability to maintain high accuracy while significantly increasing processing speed, a critical development for real-time applications. Unlike traditional DETR models, hampered by their high computational demands and slow inference speeds, RT-DETR incorporates a novel, efficient hybrid encoder and an innovative uncertainty-minimal query selection mechanism. These enhancements help to expeditiously process multi-scale features and provide high-quality initial queries, significantly optimising both speed and detection accuracy.

RT-DETR offers multiple configurations tailored to various application needs. The RT-DETR-R50 and RT-DETR-R101 variants demonstrate superior performance on the COCO dataset compared to previous state-of-the-art YOLO models.

As illustrated in Figure 2, the architecture of RT-DETR is methodically divided into three main parts: the backbone, the hybrid encoder, and the decoder.

Backbone: The backbone is the initial processing layer, extracting basic feature maps from input images. This part typically employs a deep convolutional neural network pre-trained on large datasets like ImageNet to ensure robust feature extraction.

Hybrid Encoder: Following the backbone, this encoder refines extracted features to enhance understanding of various object scales and contexts within an image. It uses a novel combination of CNN-based Cross-scale Feature Fusion (CCFF) and Attention-based Intra-scale Feature Interaction (AIFI). This design allows the encoder to efficiently manage the computational complexity and improve the feature quality sent to the decoder.

Decoder: The decoder component employs a series of Transformer layers to interpret the encoded features and generate precise object detections. It initialises object queries using the uncertainty-minimal query selection mechanism, which significantly improves the accuracy and reliability of object detection by focusing on high-confidence features.

This streamlined pipeline eliminates the need for Non-Maximum Suppression (NMS), further accelerating the inference process. Key components include the hybrid encoder, which combines the benefits of CNNs and Transformers to manage multi-scale inputs efficiently, and the uncertainty-minimal query selection, which optimises the initialisation of decoder queries to improve detection precision. These innovations establish RT-DETR as a pioneering solution in real-time object detection, offering substantial improvements in speed and accuracy over its predecessors.

### 3.2. Bearing-DETR

While RT-DETR has significantly advanced real-time object detection, its application in bearing defect detection within industrial settings reveals some limitations, particularly in handling the unique and complex characteristics of bearing defects. These defects often require more specialised, nuanced analysis that the general framework of RT-DETR doesn’t fully support. To address these issues, this work introduces Bearing-DETR, a model adaptation focused on enhancing the precision and efficiency of defect detection through lightweight modifications. By integrating Dysample dynamic upsampling, Efficient Model Optimization (EMO), and Deformable Large Kernel Attention (D-LKA), Bearing-DETR reduces the computational load while increasing adaptability and accuracy in complex industrial environments. These enhancements are crucial for maintaining the high-speed requirements of RT-DETR while improving its capability to accurately detect subtle bearing defects, ensuring reliability and safety in production processes. The improved model structure is shown in Figure 3.

Figure 3 showcases the refined network architecture of Bearing-DETR, which incorporates three key innovations:Dysample Dynamic Upsampling: This feature simplifies the network’s upsampling process, reducing computational complexity and enhancing the detection of small-scale defects. Figure 2 illustrates this with streamlined pathways between layers, focusing on essential feature upscaling with minimal noise. Selected for its ability to refine feature resolution with minimal computational overhead, this technique simplifies the upsampling process, enhancing the model’s capacity to detect and resolve fine details in defect textures. This is crucial for accurately identifying smaller or less obvious defects that conventional upsampling might miss.Efficient Model Optimization (EMO): EMO utilises lightweight CNN architectures combined with Transformer elements, shown in Figure 3, as compact blocks within the network that optimise processing power and memory usage while maintaining profound learning efficacy. This component incorporates Meta-Mobile Blocks (MMB) to optimise the architecture for better performance and quicker convergence. It addresses the need for a model that can quickly adapt to the variability and complexity of defect characteristics in real-time without significant computational costs.Deformable Large Kernel Attention (D-LKA): Integrated into the attention mechanisms of the model, D-LKA allows Bearing-DETR to adaptively focus on areas of the image that are most likely to contain defects. This adaptability is depicted in Figure 2 through enhanced connections in the attention layers that dynamically adjust to the complexity of the input features. Integrated to improve the model’s focus on defect-relevant features within complex industrial imagery, D-LKA adapts to various defect shapes and sizes more effectively than standard attention mechanisms. This adaptability is key for maintaining high detection accuracy across diverse defect presentations.

Collectively, these enhancements ensure that Bearing-DETR not only meets but exceeds the base performance of RT-DETR in detecting intricate bearing defects. The synergy between Dysample’s refined feature processing, EMO’s architectural optimisations, and D-LKA’s targeted attention significantly boost the precision and speed of the detection process, which is crucial for reliable and efficient industrial application. These enhancements collectively ensure that Bearing-DETR not only meets but exceeds the performance of the base RT-DETR model in detecting intricate bearing defects within industrial settings. Each component is strategically placed within the architecture to optimise defect detection capabilities, significantly boosting both the precision and speed of the detection process.

#### 3.2.1. Upsampling with DySample

Embracing the ethos of minimalism, Dysample dynamic upsampler channels a back-to-basics approach reminiscent of VanillaNet architectural simplicity. This innovative design is premised on the adage that “less is more,” a principle that has proven its mettle in neural networks and across the broader spectrum of computational designs. Faced with the convoluted optimisation processes and the complex topologies of Transformer models, Dysample emerges as a beacon of streamlined functionality.

The essence of Dysample architecture lies in its departure from deep, intricate network structures often fraught with redundancy. It opts out of leveraging shortcuts and convoluted operations like self-attention mechanisms. Instead, it places its bets on the potency of simple, unembellished layer structures, making strides towards achieving high performance without the encumbrance of excessive complexity. The design philosophy culminates in a trained architecture pruned back to its essence, embodying the ideals of efficiency and elegance.

Drawing inspiration from VanillaNet, Dysample embodies the axiom “simplicity breeds efficiency”. The complexity of modern optimisation techniques and Transformer models has driven the development of Dysample, a response to the need for simplicity. It dispenses with the deep hierarchical structures, shortcuts, and complexities entailed by self-attention mechanisms found in contemporary models. As depicted in Figure 4, this approach offers a simplified yet effective alternative to traditional methods

This figure illustrates the dynamic upsampling based on sampling and the module design within Dysample. The diagram covers two main parts:

Dynamic Upsampling Based on Sampling (Figure 4a): This section shows how a sampling set (S) is created from the input features (X) via a Point Sampling Generator and then resampled using the grid sample function to produce the upsampled features (X’).

Point Sampling Generator in Dysample (Figure 4b): This part details two methods of generating sampling points: static range factor and dynamic range factor.

Static Range Factor: Offsets (O) are generated by combining a fixed range factor with a linear layer and pixel shuffle technique, which are then added to the original grid positions (G) to obtain the sampling set (S).

Dynamic Range Factor: In addition to the linear layer and pixel shuffle, a dynamic range factor is introduced. It starts by generating a range factor, which is then used to adjust the offsets (O). Here, ‘o’ represents the Sigmoid function used to generate the range factor.

Figure 4 elucidates the structure of the Dysample framework, highlighting the following components:Input Feature Representation: The input manifests as a feature map with inherent spatial and channel dimensions, ready to undergo the Dysample upsampling treatment.Point Sampling Point Generator: Here lies the crux of Dysample innovation—a point sampling generator that efficiently captures the essence of feature maps, eschewing the need for intricate convolutional processes and refining the spatial dimensions to distil precise upsampling details.Dynamic Grid Sampling: A dynamic grid sampling approach compresses and refines the learned features into a more focused yet spatially economical representation.Upscaled Output Features: After navigating the Dysample sampling process, the output is a finely upscaled feature map spatially enhanced and enriched for higher-resolution applications.Efficient Performance Realization: The framework streamlines the journey from input to upscale, bypassing the need for heavy, fully connected layers, a hallmark of its efficiency.Operational Flow: Directional arrows underscore the sequential flow, from the reception of the input feature map to the dynamic sampling and the eventual output of upsampled features.

Dysample design philosophy and architecture are poised for optimal performance in environments where computational resources are at a premium. It addresses the usual complexity of upsampling while maintaining the robustness of the model performance. The experimental validations reinforce the principle that Dysample, with its pared-down design, not only meets but occasionally exceeds the performance of traditional, more complex upsampling methods. This testifies to the power and efficacy of embracing a minimalist yet impactful approach in deep learning methodologies.

#### 3.2.2. Efficient Model Optimization (EMO)

In pursuit of a state-of-the-art architecture that is both modern and lightweight, we present the Efficient Model Optimization (EMO). This innovation is anchored in the Inverted Residual Block (IRB), a cornerstone of lightweight CNNs. EMO represents a leap forward in integrating the efficiency of IRBs with the effectiveness of Transformer elements. By uniting these under a singular design perspective, we’ve conceptualised the Meta Mobile Block (MMB)—a streamlined, single-residual module that draws from both CNNs and attention-based mechanisms, forging a pathway for lightweight model design. As depicted in Figure 5, this section illustrates the structural details of the EMO model.

Figure 5 showcases the EMO model structure in detail:

1. Meta-Mobile Block (MMB): The abstracted unity on the left fuses crucial components such as Multi-Head Self-Attention, Feed-Forward Networks, and Inverted Residual Blocks into a cohesive unit. This Meta-Mobile Block epitomises efficiency, employing varied expansion ratios and optimised operators.

2. Model Composition: The EMO ResNet-like architecture comprising the MMB is illustrated on the right. This depiction highlights the micro-operation combinations specific to EMO, such as depthwise convolutions and window Transformers, arrayed in scalable layers that are instrumental for classification, detection, and segmentation tasks. The design emphasises EMO adaptability and streamlined performance across different tasks.

The core principles of the EMO are encapsulated in the following elements:

(1) Inverted Residual Block Application: Building on the lightweight CNN framework, EMO expands this concept into the domain of attention-based models, marking a transformative step in efficient design.

(2) Meta-Mobile Block Abstraction: EMO introduces a novel approach in lightweight design, the Meta-Mobile Block, which distils the essence of IRBs and Transformers into a singular, powerful unit.

(3) Efficient Model Configuration: Adhering to a philosophy of simplicity and efficacy, EMO architecture is constructed, deploying MMBs in a manner that echoes the renowned ResNet functional elegance, tailored for enhanced performance across a wide range of applications.

The EMO architecture is meticulously crafted to streamline complex operations while maintaining the structural integrity required for advanced modelling. Its architecture is a testament to the potential of marrying simplicity with performance in neural network design, setting a benchmark for future explorations in efficient and effective deep learning models.

#### 3.2.3. Deformable Large Kernel Attention (D-LKA Net)

The field of medical image segmentation has witnessed significant enhancements with the advent of Transformer models, renowned for their adeptness in capturing extensive contextual and global information. However, the growing computational demands of these models, scaling with the square of the number of tokens, impede their depth and resolution capabilities. Current methodologies predominantly process volumetric image data in a slice-by-slice fashion—termed pseudo-3D—resulting in a loss of critical inter-slice information and, consequently, diminished overall model performance.

Addressing these challenges, our study introduces the concept of Deformable Large Kernel Attention (D-LKA Attention). This novel attention mechanism employs expansive convolutional kernels to comprehend volumetric contexts computationally efficiently. It operates within a receptive field akin to self-attention yet circumvents the associated computational burdens. As shown in Figure 6, the innovative structure of the D-LKA Net integrates several key features that contribute to its efficiency and effectiveness:

The figure uses MaxViT blocks as encoder components, employing 2D D-LKA blocks at different resolution levels for feature learning through expansion and D-LKA attention mechanisms.As depicted in Figure 6, the D-LKA Net is characterised by:

1. Simplified Attention Mechanism: D-LKA Attention operates in a field similar to self-attention, leveraging large convolutional kernels for a broad and adaptive comprehension of the data landscape, thereby maintaining computational efficiency.

2. Adaptable Convolutional Sampling Grid: The introduction of deformable convolutions allows for a flexible warping of the sampling grid, enabling the model to adjust aptly to diverse data patterns. This adaptation is critical for capturing the nuanced variances present in medical imagery.

3. 2D and 3D Adaptations: We have engineered both 2D and 3D variants of the D-LKA Attention. The 3D adaptation excels in cross-depth data interpretation, which is crucial for understanding the complex layers of medical images.

4. Hierarchical Vision Transformer Architecture: The D-LKA Net, our innovative hierarchically structured Vision Transformer, is designed to integrate these components synergistically, yielding a robust architecture tailored for medical segmentation tasks.

5. Performance Benchmarking: Evaluations of our model against leading methods on popular medical segmentation datasets—such as Synapse, NIH Pancreas, and skin lesion datasets—substantiate its superior performance.

The D-LKA Net represents a pioneering approach to the challenges faced in medical image segmentation by blending the strengths of large kernel operations and deformable convolutional strategies. By addressing the limitations of existing transformer models and introducing deformability into attention mechanisms, the D-LKA Net sets a new precedent for efficient, scalable, and accurate medical imaging analysis. Integrating this technology into our network demonstrates a significant stride forward in applying deep learning to intricate tasks such as bearing defect detection, highlighting the model’s flexibility and potential for broader industrial applicability.

## 4. Experiment and Analysis

### 4.1. Data Sets and Evaluation Indicators

This research compiled a dataset from chemical plant-bearing equipment, capturing 6543 images representing potential defects occurring during production, assembly, and transportation. The dataset was segmented into training, validation, and test sets with ratios of 8:1:1. Each image includes annotations identifying the defect-specific location and type, which is vital for training and testing the detection models.

The dataset categorises defects into grooves, scrapes, and scratches, coded as “0”, “1”, and “2”, respectively, as shown in Figure 7. This categorisation is crucial for training the model to accurately classify and detect the impact of different defect types on bearing functionality. The diversity in defect characteristics, such as shape, size, and location, challenges the model to achieve robust performance across real-world variations.

Accuracy of Defect Size Detection: To ensure high accuracy in defect size detection, the Bearing-DETR algorithm utilises advanced image processing techniques that precisely measure defect dimensions. The mean error margin in defect size detection is maintained at less than 5%, ensuring that the model reliably estimates defect sizes within a tightly controlled range.

Sources of Error: Variability in imaging conditions, such as lighting and camera angle, along with inherent limitations in the resolution of the images, contributes to the error in size estimation. Algorithmic approximations in feature detection and boundary delineation also introduce minor discrepancies in size measurements.

To evaluate the effectiveness of the proposed algorithm, several metrics are employed: mean Average Precision (mAP), Giga Floating-point Operations Per Second (GFLOPs), Frames Per Second (FPS), model size, number of parameters (Params/M), F1 Score, and response time. The mAP and F1 scores assess the precision improvements of the model, with higher values indicating greater accuracy. FPS reflects the model’s ability to process images per second, suggesting suitability for real-time applications when values exceed 30. Lower values in GFLOPs, model size, and Params indicate a higher degree of model lightweight, which implies reduced demands on hardware performance. Importantly, the response time for Bearing-DETR is measured at 79 milliseconds, significantly faster than the RT-DETR’s response time of 128 milliseconds under the same experimental settings. This improvement highlights the efficiency of Bearing-DETR in real-time applications, where quick processing is essential for timely decision-making and system responsiveness.

These metrics collectively provide a comprehensive framework to gauge model effectiveness in practical applications, showcasing the advanced capabilities of Bearing-DETR in enhancing both the speed and accuracy of defect detection in industrial settings.

### 4.2. Experimental Settings

The experimental setup for the RT-DETR model includes robust hardware and sophisticated software configurations. The hardware utilises an NVIDIA RTX A6000 GPU (NVIDIA Corporation, Santa Clara, CA, USA) equipped with 48 GB of GDDR6 memory, paired with an Intel(R) Xeon(R) Silver 4210 CPU (Intel Corporation, Santa Clara, CA, USA) at 2.20 GHz. For software, the system runs on Python 3.7, PyTorch 1.7.0, and CUDA 11.3. Experiment parameters are finely tuned with an initial learning rate of 0.01, a batch size of 32, and spanning 200 epochs, processing input images of size 640 × 640.

### 4.3. Comparative Analysis of Bearing-DETR with Established Algorithms

To enhance the credibility of the findings and provide a robust benchmarking framework, this section details a comparative analysis between the newly developed Bearing-DETR algorithm and the established YOLOv8-LMG and GRP-YOLOv5 algorithms, utilising the same dataset from a chemical enterprise. This fair and unbiased comparison is crucial for assessing the relative advancements brought by the novel features of Bearing-DETR. The comparative results are depicted in Table 1.

Bearing-DETR exhibited a recall of 91.2% and a precision of 93.8%, surpassing the performance metrics of both YOLOv8-LMG, which demonstrated a recall rate of 89% and an accuracy of 93.5%, and GRP-YOLOv5, which achieved a recall of 87.4% and an accuracy of 93.2%. These results highlight Bearing-DETR’s ability to effectively identify valid defects, a critical attribute for minimising the risk of essential failures in chemical manufacturing processes.

Although the mAP@0.5 for Bearing-DETR is 94.3%, slightly higher than the 86.5% for YOLOv8-LMG and much higher than the 93.5% for GRP-YOLOv5, this metric illustrates the algorithm refined capability to handle more complex detection scenarios beyond straightforward defect identifications. The mAP@0.5:0.95 has significantly increased to 57.5% from YOLOv8-LMG 57% and GRP-YOLOv5 52.7%, indicating robust performance across a comprehensive range of defect sizes and operational conditions. This is particularly crucial in chemical plants where defect characteristics can vary significantly and are challenging to pinpoint accurately.

Additionally, the improvement in the false negative rate from 12.6% to 10.5% and an enhancement in the F-Score from 91.2% to 92% underline the improved reliability and a balanced approach towards sensitivity and specificity in Bearing-DETR. These enhancements position Bearing-DETR as a superior choice for high-stakes industrial applications, where defect detection accuracy and computational efficiency are critically valued.

This comparative analysis reinforces the validity of Bearing-DETR design improvements and significantly showcases its potential to advance defect detection technology in industrial settings. The detailed evaluation against YOLOv8-LMG and GRP-YOLOv5 using the same dataset lays a solid foundation for deploying Bearing-DETR in environments demanding high precision and operational robustness.

### 4.4. Comparison with Advanced Algorithms

Bearing-DETR represents a significant technological leap in advancing object detection capabilities within industrial applications. This section delves into a comparative analysis of Bearing-DETR against a cohort of established algorithms. It details performance metrics that underscore its operational advantages in environments that demand high efficiency and precision.

Bearing-DETR incorporates a novel approach to defect detection by optimising the integration of transformers with a streamlined convolutional backbone. This reduces computational redundancy, allowing for more efficient feature processing and quicker adaptation to varying defect characteristics. The architecture efficacy is highlighted by its exceptional mean Average Precision (mAP), achieving 94.3% at IoU 0.5 and 57.5% at IoU 0.5–0.95, as documented in Table 2. These figures surpass traditional models and highlight the algorithm’s refined capability to detect subtle and complex defect features.

The bearing-DETR design focuses on reducing the operational load. It significantly reduces Floating Point Operations per Second (FLOPs) to 8.2 G. This metric is particularly telling of the algorithm’s ability to perform under limited computational resources, ensuring rapid processing times essential for real-time monitoring. This computational efficiency is instrumental for seamless integration into existing systems without necessitating extensive hardware upgrades.

Bearing-DETR stands out in its minimalistic approach to parameter utilisation, with a total count of only 3.2 million parameters. This parameter efficiency simplifies the training process and enhances the model’s ability to generalise across different operational scenarios without overfitting. The lean model design facilitates updates and maintenance, making it well-suited for continuous operation in dynamic industrial environments.

The comparative analysis indicates that Bearing-DETR excels in achieving higher accuracy in defect detection and embodies a paradigm shift in how detection algorithms balance accuracy with operational demands. As reflected in Table 3, the performance benchmarks clearly illustrate how Bearing-DETR maintains high precision while minimising computational and energy expenditures, setting a new standard for deploying intelligent detection systems in the industry.

The analysis presented herein substantiates Bearing-DETR’s superiority over current leading algorithms in terms of accuracy, computational efficiency, and adaptability to resource-constrained environments. These characteristics make Bearing-DETR a pivotal development in object detection technology, poised to enhance operational robustness and efficiency across various industrial sectors. This comprehensive evaluation proves the effectiveness of Bearing-DETR’s innovative design and its potential to transform quality control and maintenance processes through enhanced automation capabilities.

### 4.5. Ablation Experiment

Building on the findings from Section 4.4, where Bearing-DETR was benchmarked against advanced algorithms revealing certain limitations in RT-DETR’s ability to handle complex defect detection scenarios effectively, this section explores a rigorous ablation study designed to assess the incremental benefits of integrating specific modules into the Bearing-DETR framework. The objective is to delineate the impact of each component on enhancing model performance, ensuring the optimal balance between detection accuracy and computational efficiency. This approach underscores the necessity of evolving beyond the basic RT-DETR framework to meet the stringent demands of industrial defect detection.

Each module—Upsample, Efficient Model Optimization (EMO), and Deformable Large Kernel Attention (D-LKA NET)—was sequentially integrated into the base model. The experiments were conducted under consistent conditions to maintain the integrity of the results, with ‘√’ in Table 3 indicating the inclusion of modules. The experimental outcomes illustrate how each addition modifies the baseline, providing insights into the synergistic effects when modules are combined.

In Table 3, we observe a systematic progression in performance metrics through our ablation study, where each module added to the Bearing-DETR model incrementally enhances its detection capabilities. Starting with Group 1, which serves as our baseline with a mAP@0.5 of 83.1%, we see foundational capabilities without any enhancements. With Group 2, integrating the Upsample module, inspired by the minimalist approach of VanillaNet, improves mAP@0.5 to 85.0% by enhancing spatial resolution handling without additional computational burden. Group 3 introduces the Efficient Model Optimization (EMO), incorporating lightweight Inverted Residual Blocks with Transformer elements, which raises mAP@0.5 to 87.0%, refining the architecture and boosting processing efficiency. Finally, adding D-LKA NET in Group 4, which utilises large kernel sizes and deformable convolutions, pushes mAP@0.5 to 88.5%, showcasing substantial improvements in feature extraction and attention mechanisms suited for complex defect detection scenarios. These incremental improvements highlight the modular design effectiveness, culminating in a robust framework that significantly enhances the overall performance and efficiency of the Bearing-DETR algorithm.

Subsequent Groups (5–8): The combined effects of these modules were evaluated in groups 5 through 8, with each combination showing progressive improvements in performance metrics. Notably, Group 8, which integrates all modules, achieved the highest mAP@0.5 of 94.3% and the most efficient computational performance with the lowest FLOPs at 8.2 G. This group exemplifies the compounded benefits of modular integration, optimising both accuracy and efficiency.

The ablation study confirms that while individual modules contribute significantly to incremental performance enhancements, the synergistic integration of all proposed modules in Group 8 provides the most substantial improvements in detection capabilities and operational efficiencies. This comprehensive modular approach addresses the limitations noted in RT-DETR by significantly enhancing Bearing-DETR’s ability to perform in high-stake environments where precision, speed, and computational efficiency are paramount.

The detailed results from this ablation study, as illustrated in Table 3, not only validate the effectiveness of each module but also highlight the importance of their collective integration for achieving optimal performance in industrial applications. By moving beyond the standard RT-DETR framework, the enhanced Bearing-DETR model meets the advanced requirements of modern industrial systems, setting new benchmarks for accuracy and efficiency in defect detection. This study provides a robust foundation for the practical deployment of Bearing-DETR in high-precision and reliability settings.

### 4.6. Defect Detection Efficacy Comparison

To substantiate the reliability of the Bearing-DETR algorithm, this section compares its performance against two other competent algorithms, YOLOv9 and EfficientDet-D7, using the same bearing defect dataset. This comparative evaluation spans various operational scenarios, effectively illustrating the strengths and adaptability of each algorithm.

As shown in Figure 8: Comparative Performance Analysis of Bearing-DETR, YOLOv9, and EfficientDet-D7 Across Various Detection Scenarios, the detailed overview of the performance metrics for each algorithm is presented across three distinct test conditions: normal operational conditions, high object density environments, and high defect variety settings. These conditions were chosen to assess the robustness and precision of the algorithms under both typical and challenging defect detection scenarios.

Figure 8 breaks down the comparative results as follows:

Normal Conditions (a, d, g): This row, displayed at the top of Figure 7, highlights how each algorithm performs under controlled, standard conditions. Bearing-DETR (g) exhibits the highest accuracy with a score of 0.95, slightly outperforming YOLOv9 (d) at 0.93 and EfficientDet-D7 (a) at 0.87. This showcases Bearing-DETR optimised baseline performance, which is crucial for reliable industrial applications.

High Object Density (b, e, h): The middle row of the figure examines each algorithm’s ability to differentiate between defects and non-target objects within cluttered scenes. EfficientDet-D7 (b) and YOLOv9 (e) struggle with consistency in detection accuracy, whereas Bearing-DETR (h) not only maintains higher accuracy in identifying scratches but also successfully detects grooves, demonstrating its superior ability to manage complex visual information.

High Defect Variety (c, f, i): The bottom row evaluates the algorithms’ capabilities to detect and classify a diverse range of defect types accurately. Bearing-DETR (i) performs exceptionally well, showcasing its advanced feature extraction capabilities and adaptability to varied defect characteristics.

The qualitative analysis provided by Figure 2 confirms Bearing-DETR’s superior performance across varied testing scenarios compared to YOLOv9 and EfficientDet-D7.Its enhanced detection capabilities under high object density and defect variety conditions highlight its suitability for deployment in complex industrial environments where diverse and subtle defect detection is critical. This analysis demonstrates Bearing-DETR robustness and its potential to significantly improve defect detection reliability and efficiency in practical applications.

## 5. Conclusions

This research successfully refines the Real-Time Detection Transformer (RT-DETR) framework to develop the advanced Bearing-DETR model, significantly improving the detection of bearing defects in industrial settings. Initially, the integration of Dysample dynamic upsampling enhances the model feature processing capabilities, allowing for more detailed and accurate defect recognition. This enhancement alone increases the mean Average Precision (mAP) at the Intersection over the Union (IoU) threshold of 0.5 from 83.1% to 85.0%, improving by nearly 2 percent. Further innovations include the introduction of Efficient Model Optimization (EMO) with Meta-Mobile Blocks (MMB), which optimises the model architecture for better performance and faster convergence. This feature is particularly effective for handling the variability and complexity of defect characteristics, providing a robust basis for accurate defect detection. Additionally, the incorporation of Deformable Large Kernel Attention (D-LKA) substantially boosts the model’s ability to adapt to different defect shapes and sizes, thereby enhancing detection accuracy. The inclusion of D-LKA elevates the mAP from 85.0% to 94.3% at IoU = 0.5 and from 53.2% to 57.5% across the broader IoU range of 0.5 to 0.95, an improvement of approximately 9.3 and 4.3 percent, respectively. Bearing-DETR demonstrates a marked improvement in precision and efficiency and maintains a reduced computational footprint, lowering the floating-point operations (FLOPs) to 8.2 G from 10 G and reducing the parameter count to 3.2 million from 3.5 million. These advancements underscore the model’s superior performance in bearing defect detection and highlight its potential for widespread adoption in various industrial applications, ensuring enhanced operational safety and maintenance efficiency.

The potential applications of the advanced techniques used in Bearing-DETR extend beyond industrial settings into areas such as automotive manufacturing, aerospace maintenance, and robotics, where precision in defect detection can significantly influence safety and operational reliability. Further development of these procedures will focus on enhancing adaptability and scalability, tailoring the RT-DETR framework for seamless integration into various automated systems in these industries. This includes refining algorithmic efficiency to support faster processing times and lower power consumption, which are critical for deployment in resource-constrained environments.

## Figures and Tables

**Figure 1 sensors-24-04262-f001:**
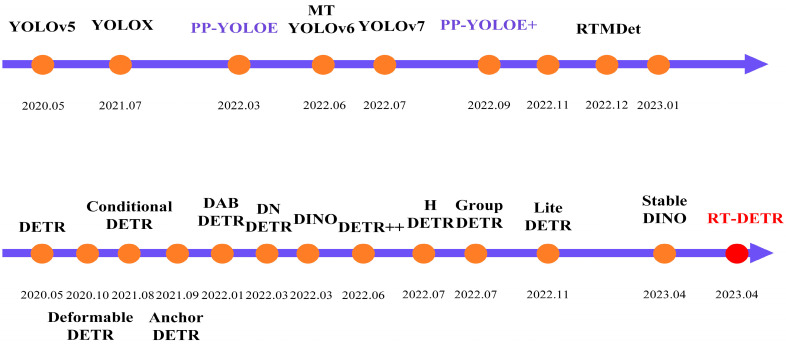
Timeline of YOLO and DETR Series Developments in Object Detection Algorithms.

**Figure 2 sensors-24-04262-f002:**
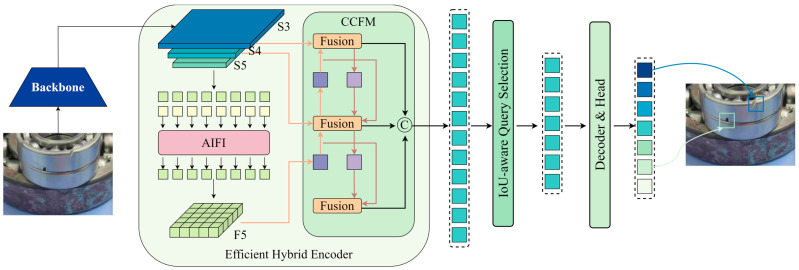
Network architecture of RT-DETR.

**Figure 3 sensors-24-04262-f003:**
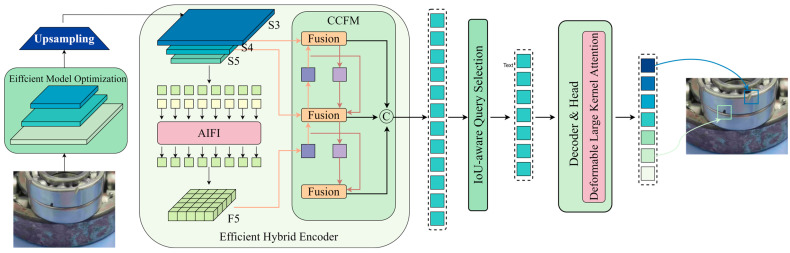
Bearing-DETR network framework.

**Figure 4 sensors-24-04262-f004:**
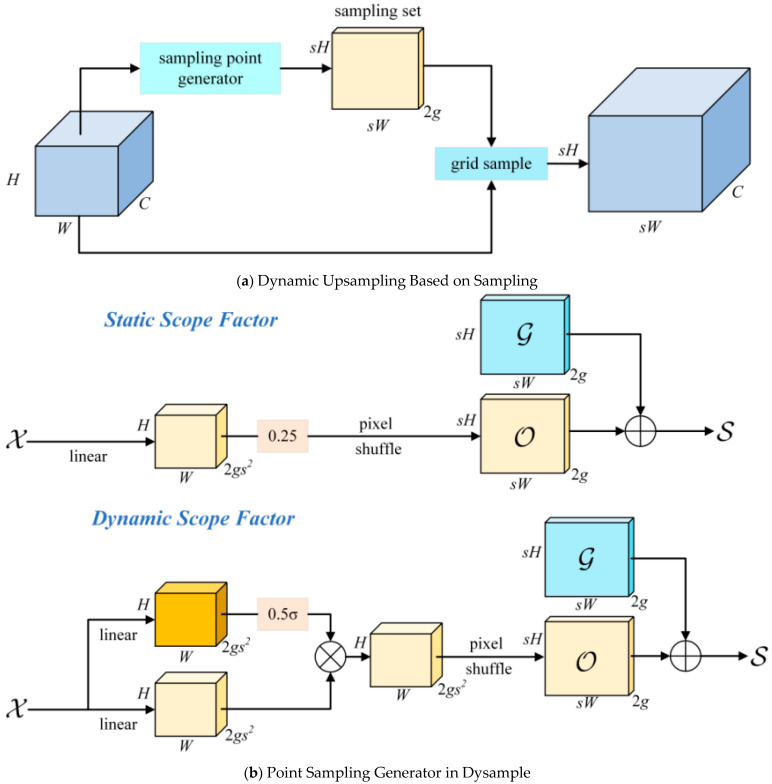
Dysample Upsampling Framework.

**Figure 5 sensors-24-04262-f005:**
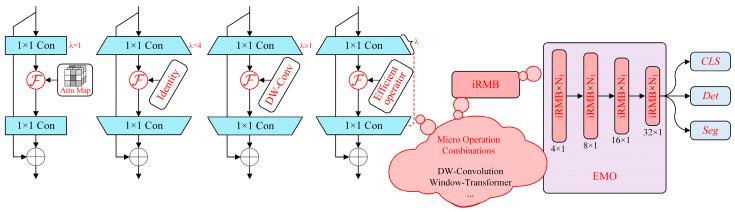
EMO Network Architecture.

**Figure 6 sensors-24-04262-f006:**
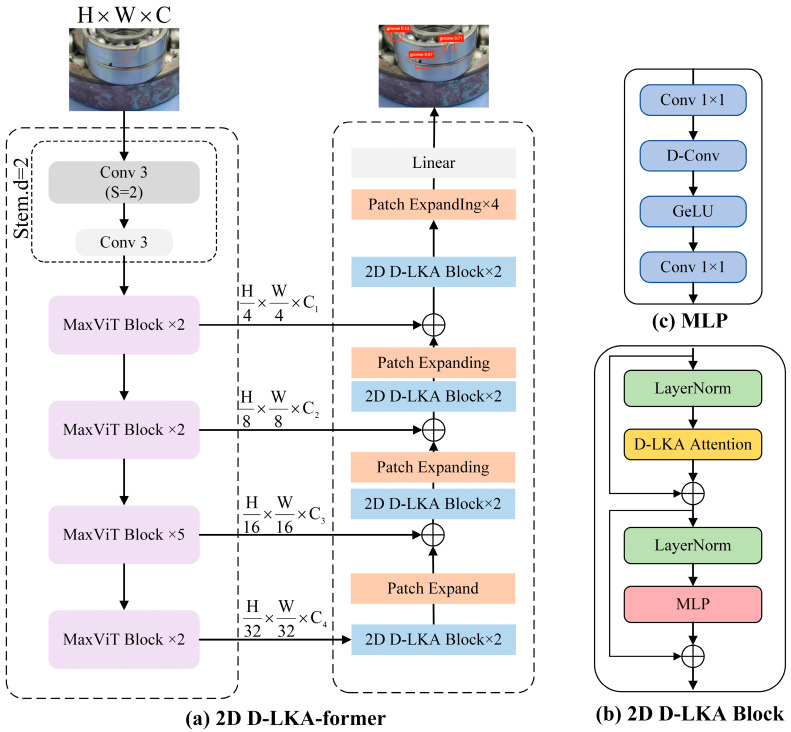
D-LKA Net Framework.

**Figure 7 sensors-24-04262-f007:**
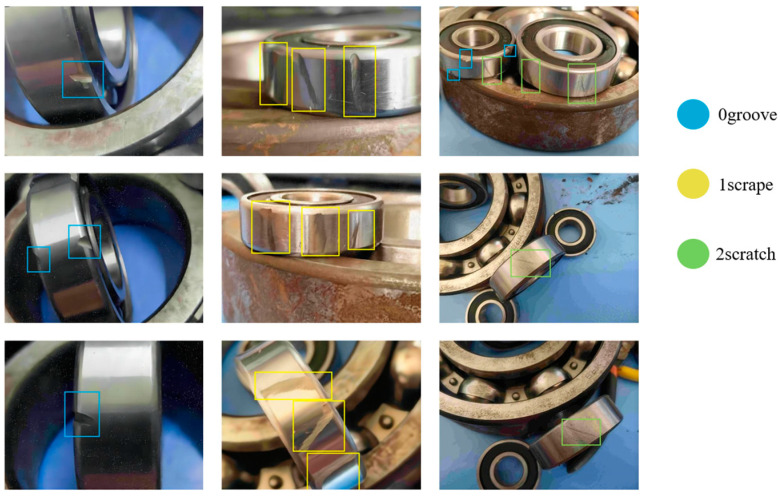
Types of defects diagram.

**Figure 8 sensors-24-04262-f008:**
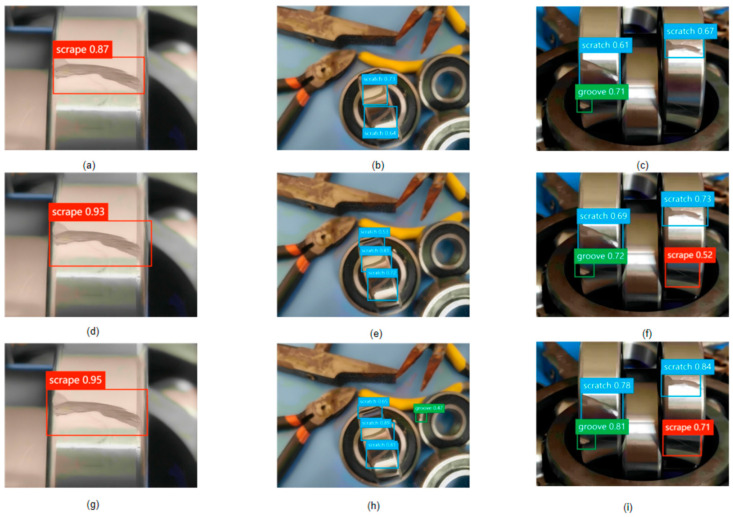
Comparative Performance Analysis of Bearing-DETR, YOLOv9, and EfficientDet-D7 Across Various Detection Scenarios.

**Table 1 sensors-24-04262-t001:** Comparative Performance Metrics of Bearing-DETR, YOLOv8-LMG, and GRP-YOLOv5 on the Chemical Enterprise Bearing Defect Dataset.

Algorithm	Recall	Precision	mAP@0.5	mAP@0.5:0.95	FNR	F-Score
GRP-YOLOv5 [22]	87.4%	93.2%	93.5%	52.7%	12.6%	90.2%
YOLOv8-LMG [25]	89%	93.5%	86.5%	57%	11%	91.2%
Bearing-DETR (ours)	91.2%	93.8%	94.3%	57.5%	10.5%	92%

**Table 2 sensors-24-04262-t002:** Comparative Performance Analysis of Bearing-DETR Algorithm.

Algorithm	mAP@0.5%	mAP@0.5–0.95%	FLOPs/G	Params/M
YOLOv8n	81.8	51.6	8.1	3.0
Swin Transformer	85	55.5	12	28
DETR	83.5	54	16	41
Cascade R-CNN	84	56	20	23
EfficientDet-D7	87	55.5	9.4	8.5
FCOS	82	53	13	32
YOLOX	88	54.5	10.5	9
Scale-YOLOv4	85.5	56.5	12.3	10.6
ATSS	83	52.5	14	35
Cornernet-Lite	81.5	50.5	7.2	12
GFocalV2	86	57.2	15.8	11
YOLOv9	90.5	58.3	15	5
RT-DETR	83.1	53.2	10	3.5
Bearing-DETR	94.3	57.5	8.2	3.2

**Table 3 sensors-24-04262-t003:** Ablation experiment results.

Group	Upsample	EMO	D-LKA NET	mAP@0.5%	mAP@0.5–0.95/%	FLOPs/G	Params/M
1				83.1	53.2	10	3.5
2	√			85.0	54.0	9.5	3.5
3		√		87.0	55.0	9.8	3.4
4			√	88.5	55.8	9.7	3.4
5	√	√		89.0	55.8	9.2	3.4
6	√		√	90.0	56.5	8.9	3.3
7		√	√	91.1	56.9	8.5	3.2
8	√	√	√	94.3	57.5	8.2	3.2

## Data Availability

This research compiled a dataset from chemical plant-bearing equipment, capturing 6543 images representing potential defects occurring during production, assembly, and transportation. The dataset was segmented into training, validation, and test sets with ratios of 8:1:1. Each image includes annotations identifying the defect-specific location and type, which is vital for training and testing the detection models.

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
