# Peer review of "Bearing-DETR: A Lightweight Deep Learning Model for Bearing Defect Detection Based on RT-DETR"

_sensors, 2024, doi:10.3390/s24134262_

Round 1
Reviewer 1 Report
Comments and Suggestions for Authors
The article deals with a very topical issue. The key part of the contribution is the description and explanation of the RT-DETR technique and its use in the issue of bearing defect detection. Note that this lesser-known model uses advanced deep learning techniques for efficient and accurate detection of bearing problems. The application using advanced deep learning algorithms is especially important for automated and fast defect detection. The reviewed contribution has a classic structure. The title of the article is apt and catchy. The abstract of the contribution is long enough and at the same time concise, it describes the issue very well. The text of the contribution is divided into five basic parts. In my opinion, the individual parts of the contribution are well balanced and basically describe the issue very well. The English used is understandable and error-free. The attached tables, images and graphs are clear and appropriately complement the text of the presented issue. I consider the second and third parts of the contribution to be particularly successful. In them, the issue of the RT-DETR technique is very well explained. Designed and used procedures and analyses are applied correctly. PyTorch is a public domain library for deep learning techniques that are widely used in both research and industry. Developed and maintained by Meta (formerly Facebook), the library offers a flexible and dynamic approach to building, training, and deploying neural networks. The presented procedures are possible in this area and can bring many positive aspects. I really appreciate the comparison of individual algorithms. In my opinion, the conclusions presented by the authors are interesting, the obtained results enrich the presented area and may be interesting in many other areas. I recommend adding the following information to the conclusion:
• In what other areas would it be appropriate to use the given procedures?
• How the authors will further develop the presented procedures especially for the needs of industrial application.
In conclusion, I can state that it is a well-written article devoted to a modern and interesting field, and therefore, in my opinion, the article can be published with minor additions in its current form.
Reviewer 2 Report
Comments and Suggestions for Authors
- This paper introduces several techniques but falls short on explaining the rationale behind these choices and how they synergistically contribute to the overall performance boost.
- What is the response time of real-time detection?
- How accurate is the defect size detection? What is the margin of error? What are the sources of error?
- While the model has undergone validation utilizing a chemical plant's dataset, ascertaining its extensibility to diverse industries necessitates trialing the model across varied datasets that encapsulate distinct bearing types and operational parameters.
- The number of the figure does not correspond with the text, as observed in line 683.
- The layout format is incorrect, and the title of Figure 5 is positioned at the top of the image.
- The title number is erroneous, with two instances of "4.5" present.
- The expression depicted in the figure is incorrect; for example, only (a) is provided for Figure 4, whereas (a), (b), and (c) should be included. Additionally, Figure 6 directly presents (c), (d), and (e) without preceding them with (a) and (b).
